# Food and feed safety of the *Bacillus thuringiensis* derived protein Vpb4Da2, a novel protein for control of western corn rootworm

Thomas Edrington, Rong Wang⬤*, Lucas McKinnon⬤, Colton Kessenich⬤, Kimberly Hodge-Bell⬤, Wenze Li, Jianguo Tan, Gregory Brown, Cunxi Wang, Bin Li, Kara Giddings

Bayer Crop Science, Chesterfield, MO, United States of America

* rong.wang@bayer.com

**Data Availability Statement:** All relevant data or URLs are within the manuscript and its Supporting Information files.

## Abstract

Western corn rootworm (WCR), Diabrotica virgifera virgifera, LeConte, is an insect pest that poses a significant threat to the productivity of modern agriculture, causing significant economic and crop losses. The development of genetically modified (GM) crops expressing one or more proteins that confer tolerance to specific insect pests, such as WCR, was a historic breakthrough in agricultural biotechnology and continues to serve as an invaluable tool in pest management. Despite this, evolving resistance to existing insect control proteins expressed in current generation GM crops requires continued identification of new proteins with distinct modes of action while retaining targeted insecticidal efficacy. GM crops expressing insecticidal proteins must undergo extensive safety assessments prior to commercialization to ensure that they pose no increased risk to the health of humans or other animals relative to their non-GM conventional counterparts. As part of these safety evaluations, a weight of evidence approach is utilized to assess the safety of the expressed insecticidal proteins to evaluate any potential risk in the context of dietary exposure. This study describes the food and feed safety assessment of Vpb4Da2, a new *Bacillus thuringiensis* insecticidal protein that confers *in planta* tolerance to WCR. Vpb4Da2 exhibits structural and functional similarities to other insect control proteins expressed in commercialized GM crops. In addition, the lack of homology to known toxins or allergens, a lack of acute toxicity in mice, inactivation by conditions commonly experienced in the human gut or during cooking/food processing, and the extremely low expected dietary exposure to Vpb4Da2 provide a substantial weight of evidence to demonstrate that the Vpb4Da2 protein poses no indication of a risk to the health of humans or other animals.

## Introduction

Demonstrating the safety of genetically modified (GM) crops for consumption by humans or other animals is a critical step towards deregulation for commercialization and consumer confidence. Biotechnology developers and global regulatory agencies have gained significant experience in assessing GM crop safety since their introduction over 25 years ago [1, 2]. The

**Funding:** The authors are employees of Bayer CropScience, a leading manufacturer of crop seeds developed through conventional breeding or biotechnology. The funder had no role in study design, data collection and analysis, decision to publish, or preparation of the manuscript. This does not alter our adherence to PLOS ONE policies in sharing data and materials.

**Competing interests:** The authors have no competing interests to declare.

successful demonstration of GM crop safety and global regulatory acceptance of new GM crop varieties has led to increased global adoption of GM crops as pivotal tools for increasing agricultural productivity amid an ever-increasing human population, which is reflected by the cultivation of GM crops on over 470 million acres across 29 countries in 2019 [3]. In particular, proteins introduced in GM crops that confer insect resistance have empowered farmers worldwide towards combatting the loss in crop productivity due to insect pest damage [4–6].

The majority of newly expressed proteins (NEPs) that convey insect resistance to GM crops have derived from the natural occurring bacterium *Bacillus thuringiensis* (*Bt*) [7, 8]. The well-established safety of *Bt*-derived insect control proteins has led to their wide-spread application through *Bt* biopesticide formulations and through incorporation into GM crops [4, 9]. The most utilized and well-characterized *Bt* proteins are insecticidal crystal (Cry) proteins that are natively expressed in *Bt* as crystalline parasporal inclusions. Cry proteins are α-pore forming proteins (α-PFPs), so named because of the α-helical structure of the final transmembrane pore, have a common three-domain architecture and share the same general mechanism of action, i.e., once ingested and exposed to the alkaline conditions on the insect midgut, the protein is activated by midgut proteases, binds specially to receptors and oligomerizes on the brush border membrane of the insect gut epithelium before undergoing a conformation change that leads to membrane insertion, pore formation and ultimately insect death [8, 10].

Cry proteins currently present in commercialized GM crops that have passed regulatory safety assessments include the Cry1 and Cry2 proteins, which provide control against lepidopteran pests, and Cry3 proteins that provide control of lepidopteran and coleopteran (Cry3) pests. A second family of insecticidal *Bt* proteins are the vegetative insecticidal proteins (Vips), which are secreted by the bacterium into the culture medium during vegetative growth [11]. The Vip3A protein is an example of a Vip protein present in commercialized GM crops that has passed regulatory safety assessments and provides control of lepidopteran pests. Vip proteins have a different overall fold as compared to Cry proteins and include members of the β-pore forming protein (β-PFPs) family, so named because they form β-barrel transmembrane pores [11–14]. Representative β-PFPs present in commercialized GM crops that have passed regulatory safety assessments include the Toxin 10 β-PFP family members Tpp35Ab1 and Gpp34Ab1 (formerly designated as Cry35Ab1 and Cry34Ab1), which act together to provide protection against western corn rootworm (WCR), Diabrotica virgifera virgifera, LeConte, and the ETX_MTX2 β-PFP family member Mpp51 (formerly designated as Cry51), which provides protection against *Lygus* and thrips species in cotton [15–18]. While Cry proteins and Vip proteins have distinct structural properties, there is growing evidence that they function through the same basic mechanism of action [11, 12, 15, 19, 20]. For example, members of the Vip3A protein family undergo processing in the target pest midgut followed by oligomerization, receptor binding, and pore formation in the gut epithelium [11]. Specifically, Vip3A proteins produce trypsin resistant polypeptides as is typically observed in processing of three-domain Cry proteins [11, 21].

The discovery and subsequent structural and functional characterization of a new *Bt*-derived insecticidal protein, Vpb4Da2 (formerly designated Vip4Da2), was recently reported [20, 22]. Vpb4Da2 exhibits high efficacy against WCR and thus provides a new tool for farmers in combatting the growing resistance of WCR to the current insect toxins present in commercial GM maize products, which include the Tpp35Ab1/Gpp34Ab1, Cry3Bb1, mCry3A and eCry3.1Ab proteins [22, 23]. The high-resolution crystal structure of Vpb4Da2 reveals a β-strand rich, six domain architecture, and combined with amino acid analysis places the Vpb4Da2 as a newly identified β-PFP with homology to the bacterial_exotoxin_B class of AB protein toxins [20, 24]. Mechanistic evaluations of the Vpb4Da2 protein *in vitro* and *in vivo*

further indicate that it follows a similar molecular mechanism of action as Cry proteins while representing a divergent and unique structural class [20].

To ensure that Vpb4Da2 can be safely consumed as part of food or feed, the safety of Vpb4Da2 for consumption by humans or other animals was evaluated herein following the Food and Agriculture Organization of the United Nations (FAO)/ the World Health Organization (WHO) international guidelines [25]. The weight-of-evidence for safety provided by these studies indicates that the Vpb4Da2 protein can be safely consumed by humans or other animals as part of feed or food.

## Methods

### Expression and purification of Vpb4Da2

The plant-produced Vpb4Da2 protein was purified from ~80 kg GM maize. Due to the low expression of Vpb4Da2 in maize seed, multiple production batches were carried out to yield enough protein for the equivalency assessment. Briefly, the ground maize powder was extracted in alkaline (pH ~11.3) buffer containing 50 mM sodium carbonate, 60 mM NaCl, 2 mM EDTA, and protease inhibitors and the resulting extract was clarified by centrifugation or filtration (ErtelAlsop) depending on the production batch. The clarified extract was loaded onto a Q Sepharose Fast Flow (Cytiva) column at pH ~10.0 and eluted using an increasing salt gradient from 0–1 M NaCl. The eluents containing the Vpb4Da2 were buffer exchanged by continuous diafiltration using a hollow fiber cartridge (Cytiva), concentrated and further purified by immunoaffinity chromatography (IAC) using a Vpb4Da2 monoclonal antibody (mAb), which was sourced by Bayer Research and Development. The final pooled protein sample was then concentrated and buffer exchanged into 10 mM sodium carbonate and sodium bicarbonate, pH ~10.0 and stored at -80˚C until use.

For recombinant Vpb4Da2 production, the coding sequence of the Vpb4Da2 protein [22] was amplified by PCR, ligated into a pET SUMO-His (Invitrogen) vector using an ig-Fusion cloning kit (Intact Genomics), and expressed in BL21 (DE3) *E. coli* (Invitrogen). Fermentation of the transfected cells was performed in the presence of kanamycin and the fermentation product was resuspended in a neutral buffer containing 50 mM Tris-HCl pH 8.5, 400 mM NaCl, 2.5 mM DTT, 2 mM $MgCl_2$, 40 mM imidazole, protease inhibitors, and lysozyme (Sigma). The fermentation slurries were then lysed using a cell disrupter (SPX) and the soluble fractions were collected and purified by Ni-NTA chromatography (G-BioSciences and Life Technologies). The His-SUMO tag was cleaved with SUMO-protease (produced internally) and removed by a second Ni-NTA column. The resulting tag-free Vpb4Da2 protein pool was then further purified by anionic exchange chromatography (Q FF). The final pooled protein sample was concentrated by ultrafiltration, and buffer exchanged by diafiltration into 20 mM sodium carbonate and sodium bicarbonate, pH ~10.0 and stored at -80˚C until use.

### Characterization of Vpb4Da2

All methods utilized to characterize the Vpb4Da2 protein are similar to what have been previously reported [26, 27]. Briefly, the purity-corrected protein concentration of the protein sample purified from maize seed and total protein concentration of the protein sample purified from the *E. coli* fermentation product were determined using gel-based densitometry and amino acid compositional analysis, respectively. The identity of both Vpb4Da2 was confirmed by N-terminal sequence determination and peptide mass fingerprint analysis using nano liquid chromatography tandem mass spectroscopy (LC-MS/MS) [28, 29]. Purity and apparent molecular weight of Vpb4Da2 were determined using densitometric analysis of Coomassie-stained SDS–PAGE gels. For western blot analysis, each protein was subjected to SDS–PAGE

and transferred to a nitrocellulose membrane. The blot was probed with an anti-Vpb4Da2 specific monoclonal antibody. Glycosylation analysis was conducted following the ECL Glycoprotein Detection method (GE Healthcare) with transferrin as a positive control.

Vpb4Da2 was tested for insecticidal activity against WCR. The activity was measured as a 7-day $EC_{50}$ value, which is the estimated protein concentration that results in 50% reduction in body mass relative to control insects. Neonate WCR larvae ($\leq$30 h old) were used to measure the insecticidal activity. Vpb4Da2 protein was incorporated into agar-based maize CRW diet (Frontier Agricultural Services) through a series of seven dilutions ranging from 0.25 to 16 µg Vpb4Da2/ml diet. Each bioassay included a buffer control in three replicates. Larvae were allowed to feed for 7 days in an environmental chamber at 25°C and 70% relative humidity before the combined weight of the survivors was assessed. The bioassay was replicated three times on separate days, each with a separate batch of insects. SAS procedure PROC NLMIXED (SAS Institute Inc.) was used to fit the data separately for each bioassay using a 3-parameter logistic model to estimate the $EC_{50}$ value.

## Bioinformatic assessment of Vpb4Da2

The Vpb4Da2 protein sequence was screened for similarity against known allergens, toxins, and protein databases in a similar process used previously [27, 30] and as described by Codex Alimentarius (2009). The Allergen Database used (herein described as AD_2021) was the "COMprehensive Protein Allergen Resource" (COMPARE) database as provided by the Health and Environmental Sciences Institute (HESI; http://db.comparedatabase.org/) and is composed of 2,348 sequences. The toxin database (TOX_2021) used was a keyword selected subset of sequences found in the Swiss-Prot database (https://www.uniprot.org/). This keyword process is used to isolate likely toxins by utilizing the search terms "(keyword:toxin OR annotation:(type:"tissue specificity" venom)) AND reviewed:yes", followed by a subsequent counter screen to remove unlikely toxins with annotations such as "antitoxin" and/or "non-toxic". The final TOX_2021 database is composed of 7,870 sequences. The all protein (PRT_2021) database was derived from ncbi-asn1 all protein fasta data from GenBank release 241 (https://ftp.ncbi.nlm.nih.gov/ncbi-asn1/protein_fasta/; accessed 01-05-2021) and is comprised of 139,450,651 sequences.

Alignments were generated with FASTA v36.3.5d run with an E-score cutoff of 1, however, a threshold of $\leq$1e−5 ($1 \times 10^{-5}$) was used for alignment significance. This is an established threshold used previously [27, 30] and is a conservative threshold for identifying homologous proteins against databases of these sizes [31].

## Assessment of Vpb4Da2 susceptibility to pepsin and pancreatin

The susceptibility of Vpb4Da2 protein to degradation by pepsin was assessed following a standardized protocol [32]. Briefly, the purified Vpb4Da2 protein was mixed with high purity pepsin (Sigma) to a final protein-to-pepsin ratio of 1 µg total protein:10 U of pepsin. The reaction mixture tube was immediately placed in a 37±2°C water bath. Samples were removed at 0, 0.5, 2, 5, 10, 20, 30, and 60 min and were immediately quenched by the addition of 0.7 M sodium carbonate and SDS-PAGE sample loading buffer. Protein only and pepsin only experimental controls were also prepared and incubated for 60 min in a 37±2°C water bath. All resulting samples were heated at 95–100°C for 5–10 min before being analyzed by SDS-PAGE and western blot analysis.

The susceptibility of Vpb4Da2 to degradation by pancreatin was also assessed. Pancreatin (Thermo Fisher Scientific) was dissolved in 50 mM potassium phosphate buffer (pH 7.5) to a concentration of 10 mg of pancreatin powder/ml as described in the United States

Pharmacopoeia [33, 34]. The pancreatin solution reaction was formulated so that 55.3 μg of pancreatin powder would be present per μg of Vpb4Da2. Samples were removed at 0, 5, 15, 30, min and 1, 2, 4, 8, and 24 h, and quenched with SDS-PAGE loading buffer before being analyzed by western blot analysis.

Limit of detection (LOD) of SDS-PAGE or western blot were also assessed using zero-minute samples for pepsin and pancreatin digestion.

### Stability of the Vpb4Da2 protein at temperatures encountered in cooking and processing

The heat stability of Vpb4Da2 was tested in separate experiments that each followed the same procedure. In each experiment, the test protein was exposed to five temperatures ranging from 25˚C to 95˚C for 15 min and 30 min, with a control sample incubated at 0˚C. Samples were placed on wet ice following incubation. A control sample aliquot of each protein was also maintained on wet ice throughout the course of the heat treatment incubation period. Each heat-treated sample was used to test for functional stability using SDS-PAGE as well as a WCR bioassay following a similar method as described above in characterization with slight modification that included a concentration range of 0.5–32 μg of Vpb4Da2.

### Acute oral toxicity assessment of Vpb4Da2

An acute oral toxicity study with Vpb4Da2 was conducted in CD-1 mice (Charles River Laboratories). The Vpb4Da2 dose solution was formulated in 10 mM carbonate/bicarbonate buffer, pH 10.0 (vehicle buffer) at dose concentration of 77.4 mg Vpb4Da2 protein per ml to enable a targeted dose level of 5,000 mg Vpb4Da2 protein/kg body weight. In addition to the test dosing solution, bovine serum albumin (BSA) was formulated in vehicle buffer (6.7 mM carbonate and bicarbonate, pH 10.0) to a concentration of 83.0 mg/ml as a protein control. Stability of the proteins over the course of dosing was confirmed by SDS-PAGE analysis and bioactivity was confirmed using an insect bioassay (as described above).

Two dose groups of ten male and ten female mice, at approximately 8 weeks of age at dosing, were administered Vpb4Da2 and BSA proteins. A 14-day observation period was conducted following dosing. Endpoints evaluated during the dosing and observation periods included: survival, clinical observations, body weights, and food consumption. Surviving animals were euthanized and subjected to a complete necropsy examination, including evaluation of the carcass and musculoskeletal system, all external orifices and surfaces, cranial cavity and external brain surfaces, and the abdominal, pelvic, and thoracic cavities and their associated organs and tissues.

Statistical analysis was conducted on body weight, body weight changes, and food consumption data obtained in the acute toxicology study conducted with Vpb4Da2. Data were subjected to a statistical decision tree. Levene's test was used to assess the homogeneity of group variances parametric assumption at the 5% significance level. The control group vs. the test dose group pairwise comparisons were conducted using a two-sided Dunnett's or Dunn's test, respectively, if the overall test was significant. Datasets with two groups were compared using a two-sided t-test or Wilcoxon Rank-Sum test, respectively. All significant pairwise comparisons were reported at the 0.1, 1, and 5% significance levels.

## Results

### Characterization and equivalency assessments of Vpb4Da2

The Vpb4Da2 protein was purified from both GM maize seed and *E. coli* cell paste (fermentation product) as described in the Methods section. Because the expression level of Vpb4Da2 is

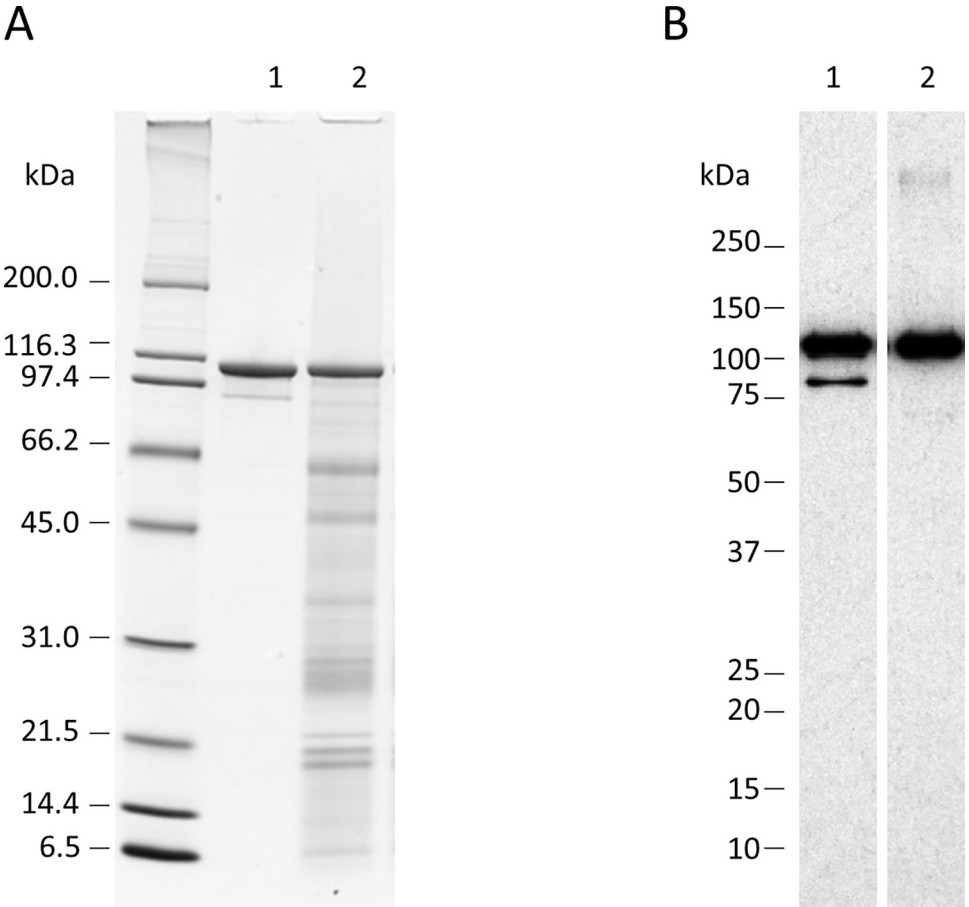

**Fig 1. Purity, molecular weight and western blot analysis of maize and *E. coli* produced Vpb4Da2.** Purified Vpb4Da2 expressed in maize and *E. coli* were subjected to precast Tris-glycine 4–20% (w/v) SDS-PAGE and western blot analysis to determine the purity, MW and immunoreactivity of each protein. **(A)** Representative SDS-PAGE gel stained with Brilliant Blue G-Colloidal stain of the *E. coli* protein (Lane 1; ~1.0 μg total protein) and maize protein (Lane 2, ~0.35 μg of Vpb4Da after purity-correction). Approximate MWs (kDa) are shown on the left. **(B)** Representative western blot after electrotransfer of *E. coli* protein (Lane 1; ~10 ng of Vpb4Da2) and maize protein (Lane 2, ~10 ng of Vpb4Da2) to a nitrocellulose membrane. Proteins were detected using an anti-Vpb4Da2 monoclonal antibody and visualized using HRP-conjugated secondary antibodies and an ECL system. Approximate MWs (kDa) are shown on the left.

very low (~1.2 ppm) in maize seeds, it was necessary to design an *E. coli* expression system to generate enough protein material to enable safety testing. Following purification of Vpb4Da2 from maize seed and *E. coli* fermentation product, characterization of the proteins was conducted followed by an assessment of their physiochemical and functional equivalency, or suitable surrogacy [26, 27, 30, 35].

The purity, molecular weight (MW), and identity of Vpb4Da2 isolated from maize seed and *E. coli* were determined (Fig 1, Table 1). In this analysis, the full-length Vpb4Da2 migrated to the same position on the SDS-PAGE gel (Fig 1, Panel A), and both proteins had an apparent MW of ~104 kDa. Western blot analysis revealed a single protein band in each sample at the same apparent MW with a minor additional protein band with an apparent MW of ~91 kDa in the *E. coli* produced sample (Fig 1, Panel B, Lane 1). Purity of Vpb4Da2 was determined to be 100% and 26% for the *E. coli*, and maize purified samples, respectively.

**Table 1. Characterization summary of Vpb4Da.**

| Characteristics | Method | Vpb4Da | |
| --- | --- | --- | --- |
| | | **E.coli–produced** | **Plant-produced** |
| Percentage Purity (%) | SDS-PAGE/Densitometry | 98 | 26 |
| Apparent MW (kDa) | SDS-PAGE/Densitometry | 103.8 | 104.9 |
| Identity | N-terminal sequence | MQNIVSSKSEQATVI | MQNIVSSKSEQATVI |
| | LC-MS/MS | 99% coverage of expected sequence | 98% % coverage of expected sequence |
| Activity ($EC_{50}$, μg protein/ml diet) | Insect Bioassay | 6.1 | 12.3 |
| Immunoreaction | western blot | Confirmed | Confirmed |
| Glycosylation | GE Glycoprotein Detection Module | none | none |

The results for glycoprotein detection are presented in Fig 2. A clear signal was observed at an apparent MW (~80 kDa) for the transferrin positive control (Fig 2, Panel A, Lanes 1 and 2). No glycosylation signals were observed for either the maize (Fig 2, Panel A, Lanes 7 and 8). or *E. coli* expressed (Fig 2, Panel A, Lanes 4 and 5) Vpb4Da2. A second membrane produced in parallel under the same conditions was stained by Coomassie R-250 (Fig 2, Panel B). Visible proteins bands were present in all lanes loaded with protein samples.

The insecticidal activity of the maize and *E. coli* expressed Vpb4Da2 was assessed and compared using a CRW diet-incorporated insect bioassay. Dose-response assays were conducted for both proteins in parallel and the assays were conducted on three separate days to estimate the mean $EC_{50}$ value (Fig 3). The mean $EC_{50}$ of the Vpb4Da2 expressed and purified from maize and *E. coli* were determined to be 12.3 and 6.1 μg protein/ml diet, respectively. Comparable insecticidal activity was observed for both proteins.

These results established the equivalency, or suitable surrogacy between the maize-expressed and *E. coli*-expressed Vpb4Da2 (Table 1). Therefore, the *E. coli*-expressed Vpb4Da2 was used for all additional safety testing.

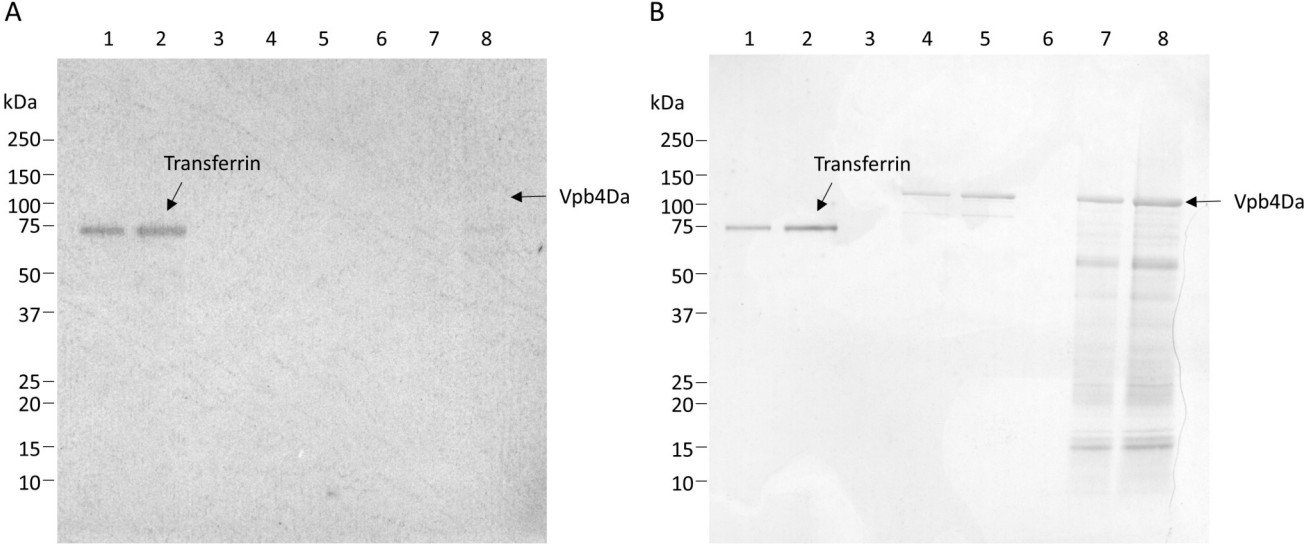

**Fig 2. Glycosylation analysis of maize and *E. coli* produced Vpb4Da2.** Purified Vpb4Da2 expressed in maize (Panels A and B, Lanes 7 and 8, ~100 ng and 200 ng, respectively) and *E. coli* (**Panels A and B**, Lanes 4 and 5, ~100 ng and 200 ng, respectively), and transferrin (positive control, Panels A and B, Lanes 1 and 2) were subjected to precast Tris-glycine 4–20% (w/v) SDS-PAGE and electrotransferred to a PVDF membrane. (A) Carbohydrate moieties were detected by addition of streptavidin conjugated HRP followed by luminol-based detection using ECL reagents and exposure to Hyperfilm® for 3 min. (B) An equivalent blot stained with Coomassie Blue R-250 to confirm the presence of proteins.

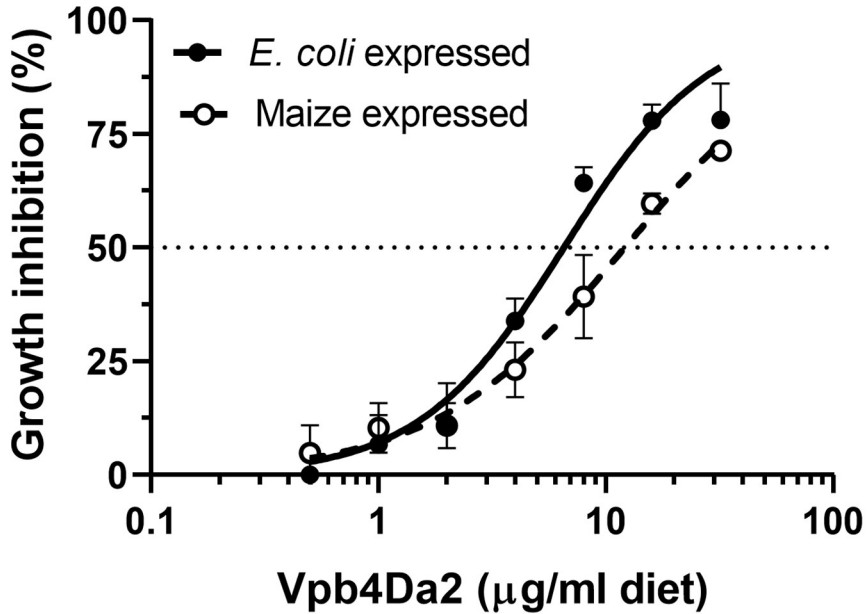

| Vpb4Da2 Protein | Mean EC$_{50}$ (ug/ml diet) | Standard Error | N |
|---|---|---|---|
| *E. Coli* expressed | 6.1 | 0.25 | 3 |
| Maize expressed | 12.3 | 0.88 | 3 |

**Fig 3. Insecticidal activity of the maize and *E. coli* expressed Vpb4Da2.** The insecticidal activities (EC$_{50}$ values) were determined from the concentration-response curves for growth inhibition of the purified Vpb4Da2 expressed in maize (dashed line) and *E. coli* (solid line) in 7-day diet incorporation bioassays. Data points represent mean values with standard errors of three replicated bioassays. The concentration-response curves were plotted with log-transformation of concentrations (X-abscissa) against the percent of growth inhibition (Y-abscissa). The mean EC$_{50}$ value and its standard error for each Vpb4Da2 were calculated from three EC$_{50}$ values estimated from three independent bioassays, respectively.

### Bioinformatic comparison of Vpb4Da2 to known allergens and toxins

The bioinformatic analyses conducted with Vpb4Da2 against the allergen database AD_2021 revealed no alignments to sequences that exceed WHO/FAO Codex Alimentarius significance thresholds, which are an *E*-score of ≤1e5, no linear matches of 8 contiguous amino acids, and no alignments displaying ≥35% identity in a sliding 80 amino acid overlap (Table 2) [25].

The screen of Vpb4Da2 against the TOX_2021 database resulted in a single alignment exceeding the *E*-score threshold of ≤1e5 (Table 2). This alignment was to a sequence described as the Protective antigen (PA) protein from *Bacillus anthracis* (UniProt id P13423), with 32.7% identity over 626 amino acids.

The search of the all-protein database (PRT_2021) resulted in the self-identification of Vpb4Da2 described as "Vpb4Da2 [synthetic construct]" (QOQ37930.1). The subsequent 4 alignments (AJT49287.1, QDF27377.1, AZJ95709.1, and AXY11547.1) were all also 100% identical to the query sequence and contained annotations indicating homology to the PA14 domain and insecticidal activity.

**Table 2. Summary of alignments for FASTA, sliding window FASTA and sliding window peptide searches of the 2021 database using the Vpb4Da2 protein sequence.**

| Database | Sequence Name | Search of the 2021 Sequence Databases | | | | | | | |
|---|---|---|---|---|---|---|---|---|---|
| | | FASTA search | | | | | | | |
| | | 8mer Hits | 35% ID 80 aa | # Hits | Accession | Description | Identity | aa Overlap | E-value |
| AD_2021 | Vpb4Da2 | No | No | 6 | AAG44480.1 | Putative vacuolar serine protease [*Penicillium citrinum*] | 25.3 | 91 | 0.013 |
| TOX_2021 | Vpb4Da2 | N/A | | 3 | P13423 | Protective antigen OS = *Bacillus anthracis* | 32.7 | 626 | 7.2e-58 |
| PRT_2021 | Vpb4Da2 | N/A | | 1014 | QOQ37930.1 | Vpb4Da2 [synthetic construct] | 100.0 | 937 | 0 |

## Heat treatment of Vpb4Da2 at temperatures common to cooking and processing

The Vpb4Da2 protein was incubated at temperatures ranging from 25 to 95°C for 15 and 30 min. Following heat treatment, the activity of the proteins relative to a control sample incubated at 0°C was assessed by an insect diet bioassay (Table 3). The results indicate that Vpb4Da2 retained insecticidal activity after incubation at temperatures up to 37°C. When incubated at temperatures of 55°C or greater for 15 min, no $EC_{50}$ value of Vpb4Da2 could be determined, indicated as "N/A" in Table 3, due to the lack of a dose response of heat-treated Vpb4Da2 on WCR larvae.

Heat-treated Vpb4Da2 protein samples were subjected to SDS-PAGE analysis and the results are presented in Fig 4. No apparent decrease in band intensity of the ~104 kDa Vpb4Da2 protein was observed and no change in the protein banding pattern was evident when samples were heated at temperatures up to 37°C for 15 or 30 min (Fig 4, Panels A and B, Lanes 1–3). Faint additional bands at both higher and lower apparent MWs began to appear in samples heated at 55°C for 30 min (Fig 4, Panel B, Lane 4). and became slightly more prominent in samples heated at 75°C for both 15 and 30 min (Fig 4, Panels A and B, Lane 5). A decrease in the intensity of the main Vpb4Da2 band was evident in sample heated at 75°C with the most prominent loss of band intensity after 30 min (Fig 4, Panels A and B, Lane 5). Incubation of samples for 15 and 30 min at 95°C (Fig 4, Panels A and B, Lane 6) resulted in a significant loss of the Vpb4Da2 band intensity with less than 10% remaining after 30 min as compared to the 10% reference standard (Fig 4, Panels A and B, Lane 8). Additionally, a significant increase in lower molecular weight protein

**Table 3. $EC_{50}$ values and 95% confidence interval (CI) after heat treatment of Vpb4Da2 protein.**

| Temperature | $EC_{50}$ (µg Vpb4Da2/ml diet (95%CI)[2] | |
|---|---|---|
| | 15 min | 30 min |
| 0°C (control) | 10 (7.4–14) | 10 (7.4–14) |
| 25°C | 14 (11–18) | 10 (6.1–16) |
| 37°C | 14 (8.0–26) | 12 (6.4–24) |
| 55°C[1] | N/A | N/A |
| 75°C[1] | N/A | N/A |
| 95°C[1] | N/A | N/A |

[1] The $EC_{50}$ value could not estimated since <50% growth inhibition was observed at the highest test concentration.

[2] The 95% confidence intervals for the bioactivity of Vpb4Da2 after each treatment temperature/time are given in the parentheses. These confidence intervals were determined by statistical analysis of the replicates for each bioassay after each temperature/time treatment of Vpb4Da2. Each range is presented in the parenthesis.

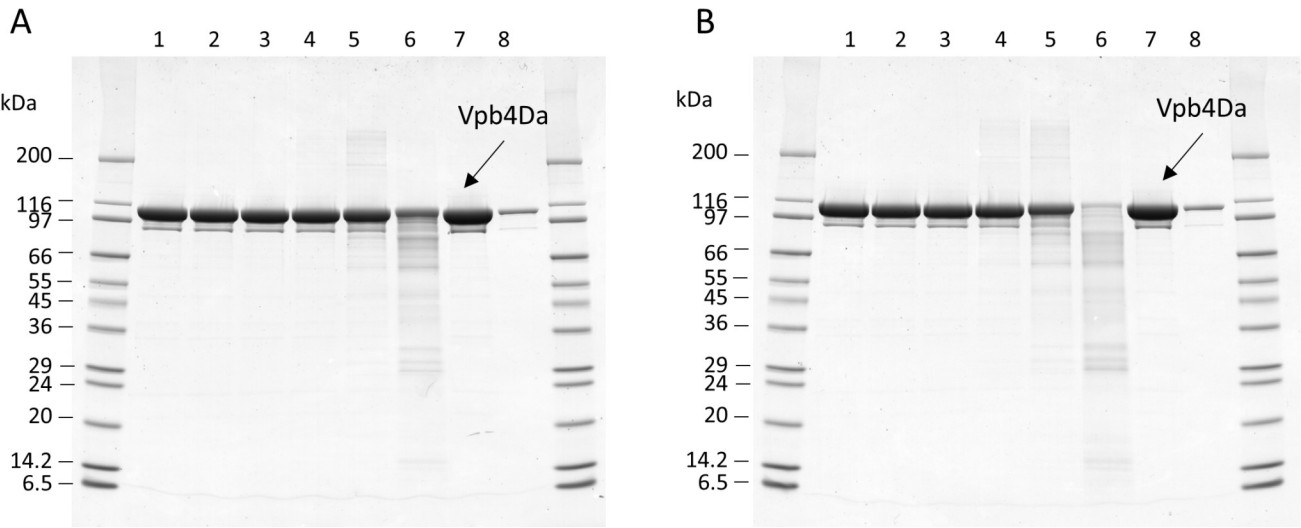

**Fig 4. Elevated temperatures lead to loss of Vpb4Da2 structural integrity.** The Vpb4Da2 protein was incubated at 0˚C (Lane 1), 25˚C (Lane 2), 37˚C (Lane 3), 55˚C (Lane 4), 75˚C (Lane 5), and 95˚C (Lane 6) for either 15 min (**Panel A**) or 30 min (**Panel B**) prior to SDS-PAGE and Coomassie staining to assess the structural integrity of the Vpb4Da2 protein at elevated temperatures. Lanes 7 and 8 are approximately 3 μg (equivalent total protein load) and 0.3 μg (10% of the total protein load) of untreated Vpb4Da2 as a control.

bands was evident in samples heated at 95˚C for both 15 and 30 min (Fig 4, Panels A and B, Lane 6).

## Susceptibility of the Vpb4Da2 to pepsin and pancreatin

The intact Vpb4Da2 was incubated with pepsin at 37±2˚C and assessed by SDS-PAGE analysis at specifically defined time intervals (Fig 5, Panel A). The intact, full-length Vpb4Da2 (~104 kDa) was degraded to below the LOD of the Coomassie-stained SDS-PAGE (Fig 5 Panel C) gel after 0.5 min, which represented >99.8% of the initial protein sample. The 2.5–6 kDa fragments were visible in the first 5 min and disappeared at 10 min (Fig 5, Panel A). There was no change in the banding pattern for the protein when incubated at 37˚C in the absence of pepsin (Fig 5, Panel A, Lanes 2 and 11). Additionally, there was no change in the protein band corresponding to pepsin (~38 kDa) when incubated at 37˚C in the absence of Vpb4Da2 (Fig 5, Panel A, Lanes 1 and 12).

The intact Vpb4Da2 was also incubated with pancreatin and assessed by western blot analysis at specifically defined time intervals (Fig 5, Panel B). Only western blot analysis was carried out due to the potential ambiguous detection from pancreatin which is a crude extract with multiple enzymes and components. The full-length protein was degraded to below the LOD of the western blot (Fig 5, Panel D) within the first 5 min (Fig 5, Panel B, Lane 4). Several bands smaller than 60 kDa corresponding to fragments of Vpb4Da2 were present during the first hour of the pancreatin digestion (Fig 5, Panel B, Lanes 4–7). Those fragments were no longer observed at 2 hours of incubation with pancreatin. In addition, such fragments would not be expected if the protein is first exposed to pepsin prior to pancreatin digestion.

## Acute toxicity testing of Vpb4Da2

An acute toxicity study with CD-1 mice was conducted using the Vpb4Da2 as the test substance and BSA as a control substance. All animals were healthy and survived to Day 14. There were no Vpb4Da2-related clinical signs observed. There were no significant differences in weekly body weights (BW) in males or females (Tables 4 and 5). Weekly body weight gains

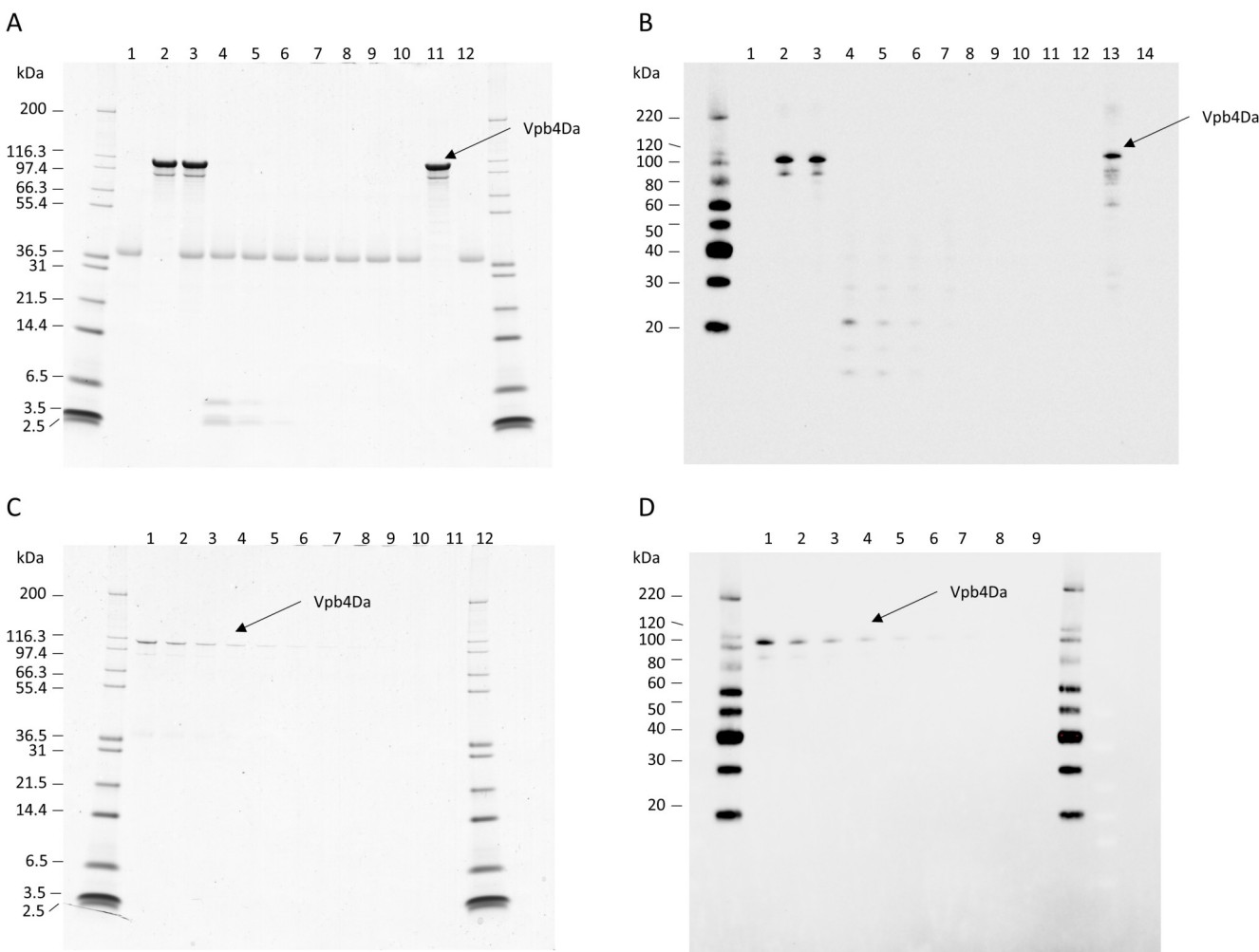

**Fig 5. Vpb4Da2 is readily degraded by gastrointestinal proteases.** Stability of the Vpb4Da2 in the presence of pepsin (Panel A) and pancreatin (Panel B) was assessed by SDS-PAGE and western blot analysis, respectively. Based on pre-reaction protein concentrations, 1 µg or 5 ng of Vpb4Da2 were loaded onto SDS-PAGE gels for electrophoresis followed by Coomassie staining or western blot analysis, respectively. SDS-PAGE gel and western blot images used to determine the LOD for the pepsin (Panel C) and pancreatin (Panel D) digestion assays are also shown. Lane designations are as follows; **Panel A**: Lane 1, pepsin only at time zero, Lane 2, Vpb4Da2 only at time zero, Lanes 3–10, time course of Vpb4Da2 incubation with pepsin for 0, 0.5, 2, 5, 10, 20, 30, and 60 min, respectively, Lane 11: Vpb4Da2 only at 60 min, Lane 12: Pepsin only at 60 min. **Panel B**: Lane 1, pancreatin only at time zero, Lane 2: Vpb4Da2 only at time zero, Lanes 3–10, time course of Vpb4Da2 digestion by pepsin at time zero, 5 min, 15 min, 30 min, 1 hr, 2 hr, 4 hr, 8 hr, and 24 hr, respectively, Lane 12, Vpb4Da2 only at 24 hr, Lane 13, pancreatin only at 24 hours. **Panel C**: Lane 1 to 11, a serial dilution from 100 ng to 0.1 ng. **Panel D**: Lane 1 to 9, a serial dilution from 2 ng to 0.008 ng.

were lower (not statistically significant by ANOVA at p<0.05) in test group males over the course of the 14-day observation period (Table 5). There was no impact of the test substance on food consumption (Table 6), and there were no abnormal gross necropsy findings present

**Table 4. Summary of acute toxicity study body weights for Vpb4Da2.**

| Sex | Mean BW (g) ± SD | | Mean BW (g) ± SD | | Mean BW (g) ± SD | |
|---|---|---|---|---|---|---|
| | Day 0 | | Day 7 | | Day 14 | |
| | Vpb4Da2 | BSA | Vpb4Da2 | BSA | Vpb4Da2 | BSA |
| Males | 36.3 ± 2.8 | 37.1 ± 3.3 | 36.6 ± 2.7 | 36.8 ± 3.4 | 37.1 ± 2.7 | 37.1 ± 3.4 |
| Females | 26.5 ± 2.3 | 27.5 ± 2.2 | 27.2 ± 2.4 | 28.2 ± 2.3 | ± 2.2 | 29.4 ± 2.9 |

**Table 5. Summary of acute toxicity study body weight changes for Vpb4Da2.**

| Sex | Mean BW Change (g) ± SD | | Mean BW Change (g) ± SD | | Mean BW Change (g) ± SD | |
|---|---|---|---|---|---|---|
| | Days 0 to 7 | | Days 7 to 14 | | Days 0 to 14 | |
| | Vpb4Da2 | BSA | Vpb4Da2 | BSA | Vpb4Da2 | BSA |
| Males | 1.0 ± 0.8 | 0.4 ± 0.9 | 0.5 ± 0.6 | 0.3 ± 0.6 | 1.5 ± 0.6 | 0.7 ± 1.3 |
| Females | 0.7 ± 1.0 | 0.8 ± 0.9 | 1. ± 0.8 | 1.1 ± 0.8 | 1.7 ± 1.3 | 1.9 ± 1.3 |

at study completion (Table 7). Vpb4Da2 did not exhibit toxicity at 5,000 mg/kg body weight (highest dose tested) in the acute toxicity study.

## Discussion

In the present study, the safety of Vpb4Da2 was assessed following guidance issued by the FAO/WHO Codex Alimentarius commission in 2009 [25] using a weight of evidence approach: 1) includes an evaluation of the history of safe use (HOSU) of Vpb4Da2 and its donor organism, *Bacillus thuringiensis*, 2) an assessment of the structural similarity between Vpb4Da2 and any known toxins or allergens, 3) characterization of the physicochemical and functional properties of Vpb4Da2, and 4) an assessment of the stability of Vpb4Da2 in the presence of gastrointestinal proteases or upon exposure to temperatures encountered during cooking and processing. Each of these parameters are typically used to evaluate the food and feed safety of all NEPs in GM crops. To further confirm the safety of Vpb4Da2 for humans and animals, an acute oral toxicity study in mice was conducted.

### History of safe use

The *vpb4Da2* gene is derived from a naturally occurring strain of *Bt*, a bacterium with an extensive HOSU dating back to 1961 [22, 36, 37]. The safety of *Bt* as part of food or feed, and by extension the proteins expressed therein has been firmly established and acknowledged by global regulatory and health agencies [38–42]. Numerous *Bt* microbial products have been directly applied to various consumed agricultural commodities globally [36, 43]. Although the exact amount of human or animal exposure to *Bt* cannot be determined, *Bt* itself has been safely consumed by humans or animals [38].

The Vpb4Da2 belongs to a class of β-PFPs found among diverse prokaryotic and eukaryotic organisms, many of which have a history of safe consumption by humans [18]. β-pore-forming Cry proteins are already present in commercialized GM crops and have been approved for food and feed use [18, 44, 45]. The Cry35 variant Cry35Ab1 (now identified as Tpp35Ab1) is present in GM maize products gained regulatory approvals in 2011 [46]. A Cry51 variant,

**Table 6. Summary of food consumption data for Vpb4Da2 acute toxicity study.**

| Group | Mean Food Consumption | | Mean Food Consumption | |
|---|---|---|---|---|
| | Days 0–7 (grams/day) | | Days 7–14 (grams/day) | |
| | Vpb4Da2 | BSA | Vpb4Da2 | BSA |
| **Males** | | | | |
| Mean ± SD | 6.70 ± 2.44 | 4.87 ± 0.59 | 6.65 ± 1.01 | 5.23 ± 0.21 |
| N | 4 | 7 | 4 | 7 |
| **Females** | | | | |
| Mean ± SD | 5.8 ± 3.14 | 5.45 ± 1.88 | 6.51 ± 2.14 | 7.15 ± 3.07 |
| N | 6 | 6 | 8 | 6 |

**Table 7. Summary of gross necropsy of Vpb4Da2 dose group.**

| Group | Dose level | Gross Pathology Observation |
|---|---|---|
| Male | 5000 mg/kg /day | No observations found |
| Female | 5000 mg/kg /day | No observations found |

mCry51Aa2 (now identified as mMpp51Aa2), is present in GM cotton and received its first regulatory approvals in 2018 [17, 45]. Although Tpp35Ab1, mMpp51Aa2, and Vpb4Da2 are active against different insect pests [17, 47], they share the same basic mechanism of action as the majority of other known insecticidal *Bt* proteins [8, 17, 20, 48, 49]. Vpb4Da2 shares the same basic mechanism of action as the majority of other known insecticidal *Bt* proteins. The MOA is consistent with well-characterized *Bt* insecticidal proteins despite structural differences; therefore, it does not exert its insecticidal activity in a completely novel manner that would be a concern for the health of humans or other animals.

## Homology to PA

The results of bioinformatic analysis using the Vpb4Da2 amino acid sequence as the query against a database of known allergen protein sequences revealed a lack of significant homology to known allergens. However, when a similar bioinformatic analysis was performed against a database of known toxin protein sequences, homology to the PA protein from *Bacillus anthracis* was observed [22]. The PA protein is one member of a three-protein exotoxin that also includes the edema factor (EF) and lethal factor (LF) proteins responsible for eliciting the toxicity of *B. anthracis* However, when a similar bioinformatic analysis was performed against a database of known toxin protein sequences, homology to the PA protein from *B. anthracis* was observed [22]. The PA protein is one member of a three-protein exotoxin that also includes the edema factor (EF) and lethal factor (LF) proteins responsible for eliciting the toxicity of *B. anthracis* [50]. Although the PA protein acts to catalyze the translocation of the two toxic protein components (i.e., EF and LF) into host cells, the PA protein itself is non-toxic and has even been safely and effectively used as a tool supporting vaccines in humans and certain animals providing protection against the *B. anthracis* infection [50, 51].

The functional and structural homology between Vpb4Da2 and other bacterial_exotoxin_B family members was conducted was previously described [20]. Kouadio et al. found that domains 1–3 of Vpb4Da2 display β-barrel architectures that are structurally homologous to domains found in a wide variety of bacterial and eukaryotic proteins including glycosyltransferases, proteases, amidases, adhesins and bacterial toxins such as the protein antigen (PA) protein from *B. anthracis* [52]. This commonly observed structural domain is named "PA14" after its location in the crystal structure of the PA protein and is typically combined in a mosaic manner with other domains involved in carbohydrate binding. In contrast, the Vpb4Da domains that confer host-specificity (i.e., domains 4–6) exhibit a low degree of sequence conservation within the same protein family [20]. The low degree of sequence conservation in Vpb4Da2 domains 4–6 suggest these domains are involved in providing specific host recognition.

A deeper look at the amino acid sequence of Vpb4Da2 reveals that an alanine is present in position 453 of the sequences, whereas a phenylalanine is present and conserved at the same aligned position in other bacterial_exotoxin_B family members [20]. This phenylalanine residue is a critical functional component of the φ-clamp of the PA protein which catalyzes the translocation of the LF and EF components across the membrane [53]. This observation

**Table 8. Example toxicity data from proteins similar to Vpb4Da2 protein.**

| Test Protein | Acute Non-Toxic Dose Level in Mice | Reference |
|---|---|---|
| mCry51Aa2 | 5000 mg/kg | [55] |
| Cry34Ab1 | 2700 mg/kg | [56] |
| Cry35Ab1 | 1850 mg/kg | [56] |
| Vip3A | >3675 mg/kg | [57] |

indicates Vpb4Da2 is highly unlikely to catalyze the translocation of other proteins into host cells following the same mechanism as observed for the PA Protein.

### *In vitro* and *in vivo* safety testing of Vpb4Da2

The characterization and equivalency results presented herein establish that *E. coli*-expressed Vpb4Da2 is a suitable surrogate for assessing the Vpb4Da2 expressed in maize. The results from *in vitro* safety testing of *E. coli*-expressed Vpb4Da2 indicate that it is unlikely that active and stable Vpb4Da2 would be present in food or feed after undergoing cooking or processing at elevated temperatures. Additionally, the results demonstrated VpbDa2 is readily degraded by gastrointestinal proteases. Therefore, Vpb4Da2 will likely be completely degraded in the human gut into its component amino acids and small peptides, which will then be absorbed and used for protein synthesis and other metabolic processes in the body [54]. The results from the *in vivo* acute toxicity assessment revealed that there was no difference in clinical end-points, behavior, and necropsy results between mice dosed with the 5000 mg of Vpb4Da2 protein/kg body weight compared to those dosed with a similar level of BSA protein. These results establish a minimal no observable adverse effect limit (NOAEL) of 5000 mg/kg body weight for the Vpb4Da2. This dose level is well above any realistic exposure level from food or feeds derived from maize expressing Vpb4Da2. Because of the low expression level of Vpb4Da2 in maize seed (~1.2 ppm), in order for a 70 kg person to receive an equivalent dose of Vpb4Da2 protein, they would have to consume ~290,000 kg (> 4000 times body weight) of maize grain in a single day, an amount that would fill ~ 290 metric tons (equivalent to what twenty large dump trucks can hold). These results are consistent with those obtained from previous studies on 3-domain Cry proteins, β-pore forming proteins, and Vip proteins conducted at high dose levels, respectively (Table 8).

### Conclusion

The Vpb4Da2 is a newly identified *Bt* protein that provides resistance to key insect pests when incorporated into GM crops. To ensure that GM crops expressing Vpb4Da2 is safe for consumption by humans or other animals, the safety of the Vpb4Da2 was assessed. The results from this assessment indicate that GM crops expressing the Vpb4Da2 pose no greater safety risk to humans or other animals than non-GM maize varieties.

### Supporting information

**S1 Raw images.**
(PDF)

**S1 File.**
(PDF)

**S2 File.**
(PDF)

## Acknowledgments

The authors would like to thank Dr. John Vicini and Dr. Kevin Glenn for their helpful comments to this manuscript. The authors would like to thank Dr. Remi Lawal for generating LC-MS/MS data.

## Author Contributions

**Conceptualization:** Thomas Edrington, Rong Wang.

**Data curation:** Rong Wang, Colton Kessenich, Kimberly Hodge-Bell, Wenze Li, Jianguo Tan, Gregory Brown.

**Methodology:** Rong Wang, Colton Kessenich, Kimberly Hodge-Bell, Wenze Li, Jianguo Tan, Gregory Brown.

**Writing – original draft:** Thomas Edrington, Rong Wang, Lucas McKinnon, Colton Kessenich.

**Writing – review & editing:** Thomas Edrington, Rong Wang, Lucas McKinnon, Colton Kessenich, Kimberly Hodge-Bell, Wenze Li, Jianguo Tan, Cunxi Wang, Bin Li, Kara Giddings.

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
