## [Decision Letter · Decision Letter 0]

24 May 2022

PONE-D-22-11120Food and feed safety of the Bacillus thuringiensis derived protein Vpb4Da2, a novel protein for control of Western Corn RootwormPLOS ONE

Dear Dr. Wang,

Thank you for submitting your manuscript to PLOS ONE. After careful consideration, we feel that it has merit but does not fully meet PLOS ONE’s publication criteria as it currently stands. Therefore, we invite you to submit a revised version of the manuscript that addresses the points raised during the review process.

Academic Editor Comments: Firstly, improve upon the English language of the MS starting from the abstract. Secondly, rectify the errors when citing the figures or the lanes in some of the gel (this is a serious issue as indicated by the reviewer 2). Thirdly, provide some background knowledge about the mode of action of VpbDa2 protein (how the protein is activated in the gut of target insect) in the introduction.    

We look forward to receiving your revised manuscript.

Kind regards,

Tushar Kanti Dutta, Ph.D.

Academic Editor

PLOS ONE

Journal Requirements:

Additional Editor Comments (if provided):

Firstly, improve upon the English language of the MS starting from the abstract. Secondly, rectify the errors when citing the figures or the lanes in some of the gel (this is a serious issue as indicated by the reviewer 2). Thirdly, provide some background knowledge about the mode of action of VpbDa2 protein (how the protein is activated in the gut of target insect) in the introduction.

Reviewers' comments:

Reviewer's Responses to Questions

**Comments to the Author**

1. Is the manuscript technically sound, and do the data support the conclusions?

Reviewer #1: Yes

Reviewer #2: Yes

2. Has the statistical analysis been performed appropriately and rigorously? 

Reviewer #1: Yes

Reviewer #2: Yes

3. Have the authors made all data underlying the findings in their manuscript fully available?

Reviewer #1: Yes

Reviewer #2: Yes

4. Is the manuscript presented in an intelligible fashion and written in standard English?

Reviewer #1: Yes

Reviewer #2: Yes

5. Review Comments to the Author

Reviewer #1: In this article titled “Food and feed safety of the Bacillus thuringiensis derived protein Vpb4Da2, a novel protein for control of Western Corn Rootworm” the authors investigated the safety of a novel insecticidal protein, Vpb4Da2 derived from Bacillus thuringiensis. Overall, this article is well composed and could be published after few modifications,

1. It will be nice to mention about the requirements for passing a protein as safe for human consumption. What are the criterion set be regulatory bodies?

2. What is the mode of action of VIPs? It would be helpful to demonstrate the similarity and differences between the structure of VIP and CRYs

3. How much maize and E.coli was used to purify the proteins?

4. What was the buffer composition used to extract Vpb4Da2 from meize?

5. Centrifugation or filtration, which method was used to clarify the maize extract

6. What was the salt gradient?

7. How buffer exchange was done?

8. I am guessing it is anti- Vpb4Da2 mAB. From where did the authors get this?

9. L104, A little bit of detail about the cloning is needed. How the coding sequence was amplified, PCR conditions, restriction enzymes etc need to be mentioned,

10. L107, what was the composition of the neutral buffer?

11. L107, what kind of cell disruptor was used? How did you get the soluble fractions?

12. L109 Typo Ni-NTA, Source of the protease?

13. Source pepsin?

14. EC50 of E. coli expressed protein was

15. What is in the parenthesis in the EC50 values in table 3

16. L315, fig 5 was referred wrong as fig 3, same in 335. But there is no fog 5B either.

Reviewer #2: The authors study whether the VpbDa2 protein which has been described to be toxic to Western corn rootworm (WCR) can be safely used in agriculture. Indeed, some other insecticidal proteins derived from Bt are commonly used to control insect pests (either as Bt biopesticide formulations or as GM crops) and passed regulatory safety assessments. In this manuscript, the authors study if the Vpb4Da2 protein sequence has any similarity against known allergens and test the acute oral toxicity in mice. Also, they assess the Vpb4Da2 protein stability to pepsin and pancreatin (two different gut proteases) and to different temperatures.

Comments:

- The first section of the results “Characterization and equivalency assessments of Vpb4Da2 proteins” can be included in the material and methods section as they are comparing the protein coming from different sources (the GMO or the recombinant E. coli strain). If the authors do not want to move at least I will suggest moving some figures and Tables to supplementary material.

- Section “Heat treatment of Vpb4Da2 at temperatures common to cooking and processing”. The authors stated that “Vpb4Da2 retained partial insecticidal activity after incubation at temperatures up to 37 °C” (see lines 286-287). However, Table 3 shows similar values (almost identical) for the proteins incubated at 0ºC, 25, or 37ºC. Please modify the sentence. Then, the authors stated: “Additional bands at both higher and lower apparent MWs began to appear in samples heated at 55°C and became more prominent in samples heated at 75°C for both 15 and 30 minutes”. However, from the point of view of this reviewer, it is hard to see these bands in the gel provided. I would suggest making softer this affirmation to be more conservative. Please, try to adjust the lane numbers in figure 4. Also, why an arrow is pointing out the band in lane 7? From my point of view, you should point the main bands in lanes 7 and 8 (both lanes contain the control protein, isn’t it?

- Section “Susceptibility of the Vpb4Da2 protein to pepsin and pancreatin”. Please carefully revised the figures that you are citing, I think you should refer to figure 5 and not to figure 3. In a similar way, please check the lanes that you want to comment on in the text because they do not match.

Why did the authors use two different methods (SDS-Page and Western Blot) to check if the protein is degraded by the two different proteases? Please comment on that.

Most of the Bt proteins need to be activated by midgut proteases to exert their toxic action on the midgut of the target insects, do you know if the Vpb4Da2 protein needs to be activated by proteases? Has this protein a resistant core to midgut proteases like Cry1 proteins? This information would be interesting either in the introduction or in the discussion.

I do not have much experience in those experiments (protein stability to pepsin and pancreatin) but is any control normally used?

- Section “Acute toxicity testing of the Vpb4Da2 protein”. I will suggest moving the sentence “These results are consistent with those obtained from previous studies on 3-domain Cry proteins, β-pore-forming proteins, and Vip proteins conducted at high dose levels, respectively (Table 7) (Lanes 349 and 350) to the discussion section, as they are not part of the results.

- As the authors organized the discussions in sections, maybe it would be possible to mix the results with the discussion to make it clearer for readers.

Minor comments:

-Line 54, remove one extra parenthesis

- Please change Vip proteins for Vip3 proteins in the text. (i.e. lines: 70, 77…)

-Line 83, correct Cry3Bb1

Lanes 86-88, please add suitable references.

Lane 105, Changer “pET SUMO-his” to pET SUMO-His

Lane 107, Please described the neutral buffer (indicate the composition)

Lane 109, Please correct the name of the column (Ni-NTA column instead No-NTA column)

6. PLOS authors have the option to publish the peer review history of their article (what does this mean?). If published, this will include your full peer review and any attached files.

Reviewer #1: No

Reviewer #2: No

---

## [Author Response · Author response to Decision Letter 0]

14 Jul 2022

Academic Editor Comments: Firstly, improve upon the English language of the MS starting from the abstract. Secondly, rectify the errors when citing the figures or the lanes in some of the gel (this is a serious issue as indicated by the reviewer 2). Thirdly, provide some background knowledge about the mode of action of VpbDa2 protein (how the protein is activated in the gut of target insect) in the introduction. 

Response: We appreciate the thoughtful comments and careful review of the entire manuscript. Changes were made throughout the manuscript to address any language issues, to make corrections, and to add clarifications.

Reviewer #1: In this article titled “Food and feed safety of the Bacillus thuringiensis derived protein Vpb4Da2, a novel protein for control of Western Corn Rootworm” the authors investigated the safety of a novel insecticidal protein, Vpb4Da2 derived from Bacillus thuringiensis. Overall, this article is well composed and could be published after few modifications,

1. It will be nice to mention about the requirements for passing a protein as safe for human consumption. What are the criterion set be regulatory bodies?

Response: Since there is no single piece of data that can solely demonstrate safety of a protein for human consumption, regulatory bodies around the world rely on a weight of evidence risk assessment when making decisions regarding the food and feed safety of newly expressed proteins (NEPs) in GM crops. A series of typical testing listed in the method section following CODEX guideline and the data presented in results section indicated no hazard potential of Vpb4Da2. As stated in the beginning of our discussion, the weight of evidence approach is based on CODEX guidelines (2009) and has been used to assess the safety of all commercialized GM crop products to date. With hazard identification as the first step, regulators want to assess the potential of a NEP to be a toxin or allergen to humans, and this is addressed through a combination of history of safe use of the NEP, history of safe use of the source organism, bioinformatics for sequence comparison to known toxins and allergens, mode of action and functional specificity of the NEP and additional in vitro and in vivo experimental work.

Risk is a function of hazard of and exposure to a given physical entity (in this case a NEP). Risk also can never be shown to be zero. This is key because risk can be determined to be low/acceptable even if one of the components (hazard or exposure) is determined to be relatively high. However, human exposure to NEPs in currently commercialized GM crops is very low due to their low expression levels in planta (ppm or ppb levels) and inactivation/degradation by digestive enzymes and high temperatures common to cooking and food processing. Furthermore, none of the NEPs in currently commercialized GM crops have been shown to have toxic or allergenic potential. 2. What is the mode of action of VIPs? It would be helpful to demonstrate the similarity and differences between the structure of VIP and CRYs

Response: Vip proteins have a different overall fold as compared to Cry proteins. They follow a similar molecular mechanism of action (MOA) as Cry proteins while representing a divergent and unique structural class. Specifically, Vip3A proteins produces trypsin resistant polypeptides as it typically occurs in three-domain Cry proteins. The proteins undergo processing in the target pest midgut followed by oligomerization, receptor binding, and pore formation in the gut epithelium. Appropriate references were provided to support the claim that Cry proteins and Vip proteins utilize the same basic mechanism of action. To elaborate, some information and references on the basic mechanism of action of a well-characterized family of Vip proteins (Vip3A) was added to the end of the third paragraph of the introduction (line 80). 

3. How much maize and E.coli was used to purify the proteins?

Response: Maize-produced Vpb4Da2 was purified from ~80 kg of seed. For the E. coli-produced protein, an amount of cell paste from fermentation was used to purify enough protein for safety studies. This information was added to the Methods section of the manuscript.

4. What was the buffer composition used to extract Vpb4Da2 from maize?

Response: The extraction buffer contained 50 mM sodium carbonate pH 11.3, 60 mM NaCl, 2 mM EDTA, and protease inhibitors. This information was added to the Methods section of the manuscript.

5. Centrifugation or filtration, which method was used to clarify the maize extract

Response: The exact method varied from batch to batch for the protein purification. However, the method of clarification does not impact the biochemical properties of Vpb4Da2 protein. This information was added to the Methods section of the manuscript.

6. What was the salt gradient?

Response: 0-1 M NaCl. This information was added to the Methods section of the manuscript.

7. How buffer exchange was done?

Response: By continuous diafiltration using a hollow fiber cartridge. This information was added to the Methods section of the manuscript.

8. I am guessing it is anti- Vpb4Da2 mAB. From where did the authors get this?

Response: The monoclonal antibody was raised against the E. coli-produced Vpb4Da2 protein and was sourced by Bayer R&D. This information was added to the Methods section of the manuscript.

9. L104, A little bit of detail about the cloning is needed. How the coding sequence was amplified, PCR conditions, restriction enzymes etc need to be mentioned,

Response: The plasmid encoding the Vpb4Da2 protein for expression in E. coli was generated by amplification of the Vpb4Da2 coding sequence by PCR followed by ligation into pET-SUMO-His using an ig-Fusion cloning kit.

10. L107, what was the composition of the neutral buffer?

Response: It contains 50 mM Tris-HCl pH 8.5, 400 mM NaCl, 2.5 mM DTT, 2 mM MgCl2, 40 mM imidazole, protease inhibitors, lysozyme, and an endonuclease. This information was added to the Methods section of the manuscript.

11. L107, what kind of cell disruptor was used? How did you get the soluble fractions?

Response: The cells were lysed by a cell disruptor from SPX. This information was added to the Methods section of the manuscript.

12. L109 Typo Ni-NTA, Source of the protease?

Response: The typo was corrected. The SUMO protease was produced and purified internally. This information was added to the Methods section of the manuscript.

13. Source pepsin?

Response: The pepsin is from Sigma. This information was added to the Methods section of the manuscript.

14. EC50 of E. coli expressed protein was

Response: 6.1 ug protein/mL diet. This information is already present in Table 1.

15. What is in the parenthesis in the EC50 values in table 3

Response: The 95% confidence intervals for the bioactivity of Vpb4Da2 after each treatment temperature/time are given in the parentheses. These confidence intervals were determined by statistical analysis of the replicates for each bioassay after each temperature/time treatment of the Vpb4Da2. This information was added to the manuscript. 

16. L315, fig 5 was referred wrong as fig 3, same in 335. But there is no fig 5B either.

Response: This mistake was corrected and the correct figure (i.e., Figure 5) was referenced in the section titled “Susceptibility of the Vpb4da2 protein to pepsin and pancreatin.” Pepsin is used to represent gastric digestion while pancreatin is used to represent intestinal digestion. Figure 5 does indeed have a panel B. Panel B shows the anti-Vpb4Da2 western blot image of the pancreatin digestion experiment samples. The legend is included in the main body of text.

Reviewer #2: The authors study whether the VpbDa2 protein which has been described to be toxic to Western corn rootworm (WCR) can be safely used in agriculture. Indeed, some other insecticidal proteins derived from Bt are commonly used to control insect pests (either as Bt biopesticide formulations or as GM crops) and passed regulatory safety assessments. In this manuscript, the authors study if the Vpb4Da2 protein sequence has any similarity against known allergens and test the acute oral toxicity in mice. Also, they assess the Vpb4Da2 protein stability to pepsin and pancreatin (two different gut proteases) and to different temperatures.

Comments:

- The first section of the results “Characterization and equivalency assessments of Vpb4Da2 proteins” can be included in the material and methods section as they are comparing the protein coming from different sources (the GMO or the recombinant E. coli strain). If the authors do not want to move at least I will suggest moving some figures and Tables to supplementary material.

Response: Establishing equivalency between the plant-produced and E. coli-produced protein is a standard practice to demonstrate that the latter can serve as a suitable surrogate in all of the later safety testing (e.g. acute toxicity, heat treatments, etc.). Presenting a summary of these experimental data in the beginning of the results is meant to provide transparency that suitable surrogate proteins were used in testing to establish the validity of the data presented in the remaining tables and figures.

- Section “Heat treatment of Vpb4Da2 at temperatures common to cooking and processing”. The authors stated that “Vpb4Da2 retained partial insecticidal activity after incubation at temperatures up to 37 °C” (see lines 286-287). However, Table 3 shows similar values (almost identical) for the proteins incubated at 0ºC, 25, or 37ºC. Please modify the sentence. Then, the authors stated: “Additional bands at both higher and lower apparent MWs began to appear in samples heated at 55°C and became more prominent in samples heated at 75°C for both 15 and 30 minutes”. However, from the point of view of this reviewer, it is hard to see these bands in the gel provided. I would suggest making softer this affirmation to be more conservative. Please, try to adjust the lane numbers in figure 4. Also, why an arrow is pointing out the band in lane 7? From my point of view, you should point the main bands in lanes 7 and 8 (both lanes contain the control protein, isn’t it?

Response: The word “partial” was removed as the insecticidal activity (EC50 values) after heat treatment at 0, 25, and 37°C was not significantly different. The statement now reads “…Vpb4Da2 retained insecticidal activity.” The higher MW bands to which we are referring are indeed very faint after 15 min at 55°C (Figure 4, Panel A, lane 4), but are more obvious after 30 min at 55°C (Figure 4, Panel B, lane 4). To avoid confusion the sentence was modified to state “…began to appear in samples heated at 55°C for 30 minutes and became…” and the exact figure panels and lanes were more clearly referenced in the text. The lane labels were adjusted, and the label was moved outside the gel image area for a clearer view of the data.

- Section “Susceptibility of the Vpb4Da2 protein to pepsin and pancreatin”. Please carefully revise the figures that you are citing, I think you should refer to figure 5 and not to figure 3. In a similar way, please check the lanes that you want to comment on in the text because they do not match.

Why did the authors use two different methods (SDS-Page and Western Blot) to check if the protein is degraded by the two different proteases? Please comment on that.

Response: The mistakes in figures and lanes referenced was corrected. Now the intended figure and lanes are referenced. Two different detection methods are used due to the difference in purity between pepsin and pancreatin. The pepsin used in the in vitro digestion experiments is a highly purified form of this enzyme. Therefore, digestion fragments of test proteins can typically be detected with Coomassie staining without interference from the pepsin protein band or any other minor bands that may come from pepsin. On the other hand, pancreatin is a mixture of enzymes and is a crude commercially available extract. Digestion fragments of test proteins mixed with pancreatin cannot be unambiguously detected with Coomassie staining because of the large number of bands present in pancreatin alone. Therefore, western blotting is used to detect test proteins and their pancreatin digestion fragments. The methods used for detection in each digestion experiment are also recognized and accepted by global regulatory agencies. One clarification sentence is added to the Results section of “Susceptibility of the Vpb4Da2 protein to pepsin and pancreatin”.

Most of the Bt proteins need to be activated by midgut proteases to exert their toxic action on the midgut of the target insects, do you know if the Vpb4Da2 protein needs to be activated by proteases? Has this protein a resistant core to midgut proteases like Cry1 proteins? This information would be interesting either in the introduction or in the discussion.

Response: Yes, the Vpb4Da2 protein is activated by proteases in the insect midgut and is processed to a stable core of ~80-kDa. This is detailed in Kouadio et al (2021, PLoS One) which we reference in the text multiple times. In the penultimate paragraph of the introduction (lines 84-93) we provide an overview of the Vpb4Da2 protein including basic structural features and a statement that the mechanism of action of this protein has been investigated and is similar to that of known Cry proteins. The appropriate reference for these data is included. We could elaborate on the details, but we feel that this would mostly repeat what is stated in the second paragraph of the introduction (lines 60-64).

I do not have much experience in those experiments (protein stability to pepsin and pancreatin) but is any control normally used?

Response: The control proteins are often used to illustrate the performance of the pepsin or pancreatin. It is not sensitive enough to inform the quality or batch-to-batch difference of the enzymes. The quantitative and reproducible approach is to verify the activity of pepsin and pancreatin prior to conducting the digestions with a fixed enzyme: test protein ratio. 

- Section “Acute toxicity testing of the Vpb4Da2 protein”. I will suggest moving the sentence “These results are consistent with those obtained from previous studies on 3-domain Cry proteins, β-pore-forming proteins, and Vip proteins conducted at high dose levels, respectively (Table 7) (Lanes 349 and 350) to the discussion section, as they are not part of the results.

Response: This sentence was moved to the end of the Discussion section as it is a more logical location for this summary statement.

- As the authors organized the discussions in sections, maybe it would be possible to mix the results with the discussion to make it clearer for readers.

Response: Including the results and discussion of the meaning/implications of those results in one large section after the Methods could be done. However, we feel the current layout most succinctly summarizes our work and is consistent with the layout of many other publications in PLoS One.

Minor comments:

-Line 54, remove one extra parenthesis

Response: This error was corrected.

- Please change Vip proteins for Vip3 proteins in the text. (i.e. lines: 70, 77…)

Response: In lines 70 and 77, we are not referring to just Vip3 proteins but Vip-class proteins more broadly. Some words were added for clarification.

-Line 83, correct Cry3Bb1

Response: This error was corrected.

Lanes 86-88, please add suitable references.

Response: A suitable reference was added

Lane 105, Change “pET SUMO-his” to pET SUMO-His

Response: This error was corrected.

Lane 107, Please describe the neutral buffer (indicate the composition)

Response: It contains 50 mM Tris-HCl pH 8.5, 400 mM NaCl, 2.5 mM DTT, 2 mM MgCl2, 40 mM imidazole, protease inhibitors, lysozyme, and an endonuclease. This information was added to the Methods section of the manuscript.

Lane 109, Please correct the name of the column (Ni-NTA column instead No-NTA column)

Response: This error was corrected.

---

## [Editor Report · Decision Letter 1]

18 Jul 2022

Food and feed safety of the Bacillus thuringiensis derived protein Vpb4Da2, a novel protein for control of Western Corn Rootworm

PONE-D-22-11120R1

Dear Dr. Wang,

We’re pleased to inform you that your manuscript has been judged scientifically suitable for publication and will be formally accepted for publication once it meets all outstanding technical requirements.

Kind regards,

Tushar Kanti Dutta, Ph.D.

Academic Editor

PLOS ONE

Additional Editor Comments (optional):

Kindly italicize Diabrotica virgifera virgifera throughout the manuscript during proof stage of the article.
---

## [Editor Report · Acceptance letter]

25 Jul 2022

PONE-D-22-11120R1 

Food and feed safety of the *Bacillus thuringiensis* derived protein Vpb4Da2, a novel protein for control of Western Corn Rootworm 

Dear Dr. Wang:

I'm pleased to inform you that your manuscript has been deemed suitable for publication in PLOS ONE. Congratulations! Your manuscript is now with our production department. 

Kind regards, 

on behalf of

Dr. Tushar Kanti Dutta 

Academic Editor

PLOS ONE